# Role of RadA and DNA Polymerases in Recombination-Associated DNA Synthesis in Hyperthermophilic Archaea

**DOI:** 10.3390/biom10071045

**Published:** 2020-07-14

**Authors:** Gaëlle Hogrel, Yang Lu, Nicolas Alexandre, Audrey Bossé, Rémi Dulermo, Sonoko Ishino, Yoshizumi Ishino, Didier Flament

**Affiliations:** 1Laboratoire de Microbiologie des Environnements Extrêmes, Ifremer, CNRS, Univ Brest, 29280 Plouzané, France; gaelle.hogrel@ifremer.fr (G.H.); yang.lu@ifremer.fr (Y.L.); alexandre.nicolas2@outlook.fr (N.A.); audrey.bosse@ifremer.fr (A.B.); remi.dulermo@ifremer.fr (R.D.); 2LIA1211 MICROBSEA, Sino-French International Laboratory of Deep-Sea Microbiology, 29280 Xiamen-Plouzané, France; 3Department of Bioscience and Biotechnology, Graduate School of Bioresource and Bioenvironmental Sciences, Kyushu University, 744 Motooka, Nishi-ku, Fukuoka, Fukuoka 819-0395, Japan; sonoko@agr.kyushu-u.ac.jp (S.I.); ishino@agr.kyushu-u.ac.jp (Y.I.)

**Keywords:** homologous recombination, Archaea, DNA polymerase, recombinase, DNA repair

## Abstract

Among the three domains of life, the process of homologous recombination (HR) plays a central role in the repair of double-strand DNA breaks and the restart of stalled replication forks. Curiously, main protein actors involved in the HR process appear to be essential for hyperthermophilic Archaea raising interesting questions about the role of HR in replication and repair strategies of those Archaea living in extreme conditions. One key actor of this process is the recombinase RadA, which allows the homologous strand search and provides a DNA substrate required for following DNA synthesis and restoring genetic information. DNA polymerase operation after the strand exchange step is unclear in Archaea. Working with *Pyrococcus abyssi* proteins, here we show that both DNA polymerases, family-B polymerase (PolB) and family-D polymerase (PolD), can take charge of processing the RadA-mediated recombination intermediates. Our results also indicate that PolD is far less efficient, as compared with PolB, to extend the invaded DNA at the displacement-loop (D-loop) substrate. These observations coincide with previous genetic analyses obtained on *Thermococcus* species showing that PolB is mainly involved in DNA repair without being essential probably because PolD could take over combined with additional partners.

## 1. Introduction

The evolution has equipped the three domains of life, Bacteria, Archaea and Eukarya, with a faithful process to repair Double-Strand Breaks (DSB): Homologous Recombination (HR). This pathway restores genetic information by copying a homologous template sequence [1]. Other DSB repair pathways operate alongside HR: the Non-Homologous End-Joining (NHEJ) [2] and Alternative End-Joining (a-EJ) pathways [3]. Efficient and less energetically demanding, these two alternative pathways are however more error-prone processes as they can result in deletions. Components involved in NHEJ and a-EJ are inexistent or partially conserved in archaeal species in comparison to Bacteria and Eukarya [4].

In Archaea, HR pathway seems to have major functions not only for DSB repair, but also to initiate replication. In this regard, several genetic studies in Crenarchaeota and Euryarchaeota highlighted the essentiality of HR proteins (such as Mre11 and RadA) in hyperthermophilic Archaea where their deletion resulted in lethal phenotypes [5,6,7]. In the last decade, studies on archaeal strains deleted from replication origins, revealed the potential role of HR proteins to initiate DNA synthesis outside canonical origins [8,9]. In *Haloferax volcanii*, the four chromosomal origins (oriCs) can be deleted together from the genome without any growth defect, but the *radA* gene is essential in the *oriC*-less strain [8]. Together these observations raised interesting questions about the role of HR in replication strategies within the archaeal domain.

During the HR process, the DSB is detected and resected by exonucleases to generate 3′ single-stranded DNA (ssDNA) tails. This ssDNA will be the starting point for the search of a homologous sequence that could be used as a template to restore the lost genetic information. For the homology search, RecA-family recombinases are key proteins. Named RecA in Bacteria, Rad51 in Eukarya and RadA in Archaea [10,11], they all bind ssDNA in a manner that leads to the formation of a nucleoprotein filament (reviewed in [12]). Structural and biophysical analysis of the filamentous forms of the recombinase along DNA have revealed that the dynamic flexibility of Rad51 nucleoprotein filament is dependent on the binding and hydrolyzing of ATP molecules ([13,14,15,16,17,18,19]). Rad51 interacts with DNA through its phosphate backbone with two binding sites used to interact both with ssDNA and dsDNA allowing homology search. When this homologous sequence has been successfully found, the DNA strand exchange between the donor sequence and the recombinase filament leads to a specific structure, known as the displacement-loop (D-loop).

In the D-loop, a structure similar to a replication fork is formed. The invasive strand brought by the recombinase provides a primer with a 3′ end available for further 5′ to 3′ DNA synthesis by DNA polymerases. In Archaea, DNA polymerase operating after the strand exchange step has not been identified. To date, DNA-dependent DNA polymerases have been classified based on their primary structures into six families: A, B, C, D, X and Y. Previous biochemical studies have shown that several DNA polymerases are capable of DNA synthesis on recombination intermediates: (i) Pol IV (Y), Pol II (B) and Pol III (C) in Bacteria [20,21,22], (ii) Pol ζ, Pol δ, Primase/Pol α (B) and Pol η (Y) in Eukarya [23,24,25,26,27,28,29]. A genetic system in yeast revealed that both replicative DNA polymerases, Pol δ and Pol ε, are required for HR [28], but the precise role of Pol ε remains to be determined [29].

In Archaea, only three DNA polymerase families have been identified: B, D and Y (see reviews [30,31]). All Archaea contain family-B polymerase (PolB) structurally similar to eukaryotic replicative DNA polymerases [32,33]. The family-D represented by PolD [34] is more unusual as it presents functional properties of a replicative DNA polymerase but with a catalytic subunit sharing similarities with RNA polymerases [35]. PolD is widely distributed among Archaea, except the crenarchaeal organisms, but is never found in Bacteria and Eukarya. Finally, the family-Y DNA polymerase has been found only in Crenarchaea phylum [36].

Our study model, the hyperthermophilic euryarchaea, *Pyrococcus abyssi*, encodes three enzymes capable of DNA synthesis: PolB, PolD and the p41/p46 complex. The p41/p46 complex is involved in RNA primer synthesis but is also a potential DNA repair enzyme [37,38,39,40]. Several biochemical characterization studies contributed to show that both PolB and PolD are processive enzymes with high nucleotide selectivity and 3′-5′ proofreading activities [32,33,34,39,41,42,43,44,45,46].

The functional role of PolB and PolD in DNA replication and/or DNA repair has been debated for a long time. Inspired by the respective role of Pol ε and Pol δ in eukaryotic DNA replication in addition to biochemical studies such as primer usage and strand-displacement activity [47,48], it has been suggested that PolB replicates the leading strand, whereas PolD replicates the lagging strand. In another side, gene deletion studies indicate that only PolD is required for DNA replication while PolB is important for DNA repair but not essential in Thermococcales [49,50,51]. What about their role in HR process in Archaea? Our understanding of the precise mechanisms by which DNA synthesis takes place during HR in Archaea is incomplete. The biochemical properties such as, high processivity, strand-displacement activity and proof-reading activity make PolB and PolD ideally suited for accurate DNA synthesis on recombination intermediates.

Working with purified *P. abyssi* proteins, here we reconstituted an in vitro assay for recombination-associated DNA synthesis. Based on previous studies with eukaryotic proteins [26,52], we used a purified supercoiled plasmid as the homologous donor substrate and a ssDNA of 93 nt as the invading strand. In the presence of the RadA recombinase, we managed to produce D-loop substrate and showed that both PolB and PolD can take over RadA-mediated recombination intermediates. Interestingly our results also indicate that PolD is less efficient as compared to PolB to extend the invaded strand in the D-loop substrate. In regards to both biochemical and genetical observations from previous publications, our results support an important role for PolB in HR but also identify PolD as a possible back-up polymerase.

## 2. Materials and Methods

### 2.1. Protein Expression and Purification

#### 2.1.1. RadA

*P. abyssi* RadA was purified following the protocol published for *P. furiosus* RadA [10]. The sequence of the gene encoding RadA (Gene ID: 1495130) was optimized for expression in *E. coli* cells and cloned into pET-30a(+) by Genecust (Ellange, Luxembourg) (optimized sequence in Appendix A). RadA was overexpressed in *E. coli* BL21 (DE3) cells carrying RadA/pET-30a(+). IPTG (isopropyl-1-thio-D-galactopyranoside) was added to a final concentration of 0.1 mM, and the cells were further grown for 4 h at 37 °C to induce the expression from RadA/pET-30a(+). The cells were harvested by centrifugation at 5000× *g* for 30 min. The cell pellet was resuspended into 50 mM Tris-HCl, pH 8.0, 200 mM NaCl, 0.1 mM EDTA, 0.5 mM DTT complemented with protease inhibitors (Roche, #05056489001, Basel, Switzerland). The cells were disrupted by pressure (One shot, 1.8 kbar) and sonication then centrifuged at 10,000× *g* for 20 min. The supernatant was incubated at 80 °C for 20 min and centrifuged again at 10,000× *g* for 20 min. Then the soluble fraction was diluted two times with buffer A: 50 mM Tris-HCl, pH 8.0; 0.1 mM EDTA; 0.5 mM DTT; 5% glycerol to decrease salt concentration before protein precipitation with 0.1% PEI (Polyethyleneimine), P3143, Sigma, Saint-Louis, MO, USA). The protein sample was incubated for 5 min on ice and was centrifuged for 10 min at 3000× *g*. The protein pellet was then resuspended with one equal volume of buffer A complemented with 0.4 M ammonium sulfate and incubated for 15 min on ice. The soluble fraction was recovered after centrifugation at 3000× *g* for 10 min and mixed with ammonium sulfate to a final concentration of 2 M before being loaded onto a phenyl-sepharose column (Hiprep phenyl (high sub) FF 16/10, GE Healthcare, Chicago, MI, USA). RadA protein was eluted at 160 mM ammonium sulfate and the fractions were dialyzed against buffer A containing 0.2 M NaCl. The dialysate was applied onto a heparin column (HiTrap heparin, GE Healthcare, Chicago, MI, USA) and eluted at 0.7–1.0 M NaCl. RadA protein was then concentrated with Sartorius Vivaspin (30,000 MWCO, Göttingen, Germany) and applied onto a gel filtration column (Superdex 200, GE Healthcare, Chicago, MI, USA) in the buffer A complemented with 0.2 M NaCl. The final RadA concentration was calculated by measuring absorbance at 280 nm with a predicted extinction coefficient of 12,950 M^−1^·cm^−1^ and molecular weight of 38,760.4 Da (ProtParam, ExPASy, [53]).

#### 2.1.2. PCNA (Proliferating Cell Nuclear Antigen)

*P. abyssi* PCNA (Gene ID: 1496768) was produced and purified as described previously [54]. Briefly, PCNA was overproduced by addition of 1 mM IPTG in *E. coli* strain BL21-CodonPlus-RIL grown for 4 h in Lysogeny broth (LB) at 37 °C. The cell extracts were treated by heating at 80 °C for 10 min, and His-tagged PCNA was purified by using a strong anion exchange column (HiPrep Q FF 16/10, GE Healthcare, Chicago, MI, USA) and an affinity column (HisPrep FF 16/10, GE Healthcare, Chicago, MI, USA). After dialysis and concentration, the final protein sample was stored in 40 mM Tris-HCl, pH 8.0; 0.8 mM DTT; 240 mM NaCl; 0.16 mM EDTA and 20% glycerol. Protein concentration was measured by colorimetric assay based on the Bradford dye-binding method (BioRad protein assay dye reagent, BioRad, Hercules, CA, USA). The recombinant *Pa*PCNA has a predicted extinction coefficient of 7450 M^−1^·cm^−1^ and a molecular weight of 29,433.77 Da (ProtParam, ExPASy, [53]).

#### 2.1.3. PolB

exo+ and exo− versions of *P. abyssi* PolB (Gene ID: GI:1495739) were produced and purified as already described [33].

Briefly, PolB exo+ and exo− were overproduced by addition of 1 mM IPTG in *E. coli* strain Rosetta 2(DE3)pLysS grown overnight in Lysogeny broth (LB) at 30 °C. The cell extracts were treated by heating for 20 min at 75 °C, and His-tagged PolB was purified by using an affinity column (HisPrep FF 16/10, GE Healthcare, Chicago, MI, USA) and by hydrophobic chromatography (HiPrep Phenyl (low sub) FF 16/10, GE Healthcare). After dialysis and concentration, the final protein sample was stored in 30 mM Tris-HCl, pH 7.5; 0.6 mM DTT; 60 mM NaCl and 40% glycerol. Protein concentration was calculated by measuring absorbance at 280 nm. The recombinant *Pa*PolB has a predicted extinction coefficient of 122,050 M^−1^·cm^−1^ and a molecular weight of 89,400.31 Da (ProtParam, ExPASy, [53]).

#### 2.1.4. PolD

exo+ and exo− versions of *P. abyssi* PolD, DP1 (small subunit, Gene ID: 1495007) and DP2 (large subunit, Gene ID: 1495008),were produced and purified as already described [44,55]. Briefly, PolD exo+ and exo− were overproduced by addition of 1 mM IPTG in *E. coli* strain grown overnight in Lysogeny broth (LB) at 20 °C. The cell extracts were treated by for 30 min at 70 °C, and his-tagged PolD was purified by using a strong anion exchange column (HiLoad 26/10 Q sepharose, GE Healthcare, Chicago, MI, USA), two affinity columns (HisPrep FF 16/10 and HiTrap Heparin, GE Healthcare, Chicago, MI, USA) and a size exclusion column (Superdex 200, GE Healthcare, Chicago, MI, USA). After concentration, the final protein sample was stored in 35 mM Tris-HCl, pH 7.5; 0.7 mM β-mercaptoethanol; 35 mM NaCl and 30% glycerol. Protein concentration was calculated by measuring absorbance at 280 nm. The recombinant *Pa*PolD exo+ has a predicted extinction coefficient of 63,720 M^−1^·cm^−1^ and a molecular weight of 69,395.36 Da for DP1; 152,990 M^−1^·cm^−1^ and 146,368.57 Da for DP2 (ProtParam, ExPASy, [53]). The recombinant *Pa*PolD exo− has a predicted extinction coefficient of 65,210 M^−1^·cm^−1^ and a molecular weight of 74,834.07 Da for DP1; 152,990 M^−1^·cm^−1^ and 144,205.25 Da for DP2 (ProtParam, ExPASy, [53]).

### 2.2. DNA Substrates

#### 2.2.1. Circular D-Loop

The L93-5′HL 647 oligonucleotide (5′HiLyte^TM^ Fluor 647-AAA-GGC-GGT-AAT-ACG-GTT-ATC-CAC-AGA-ATC-AGG-GGA-TAA-CGC-AGG-AAA-GAA-CAT-GTG-AGC-AAA-AGG-CCA-GCA-AAA-GGC-CAG-GAA-CCG-TAA-AAA-3′ [26,56] was obtained from Eurogentec (Liege, Belgium). This oligonucleotide is 100% homologous to pUC19 region 2523–2615. The supercoiled form of pUC19 was purified from *E. coli* DH5α transformed with pUC19 plasmid (TAKARA©, Kusatsu, Japan). *E. coli* cells were grown overnight at 37 °C in LB media with 100 µg/mL ampicillin. Then the cells were rapidly cooled in ice before centrifugation at 6000× *g* for 15 min. pUC19 plasmid was extracted from the cell pellet by using QIAfilter™Plasmid Maxi Kit (Qiagen, Hilden, Germany) according to the manufactural protocol. Then the different form of pUC19 plasmid were separated by electrophoresis on an 1% TAE agarose gel and visualized by ethidium bromide staining. The supercoiled DNA ladder (N0472S, NEB, Ipswich, MA, USA) was used to identify the supercoiled form of pUC19 (2686 bp) and extracted the corresponding band. The DNA was eluted from the agarose gel using phenol (77607, Sigma, Saint-Louis, MO, USA) then the sample was frozen for 5 min at −80 °C and finally centrifuged at 12,000× *g* for 15 min. The plasmid was recovered from the supernatant phase and precipitated with 1:10 volume of 5 M NaCl and 2.5 volume of cold ethanol and incubated for 15 min at −80 °C. After centrifugation for 15 min at 12,000× *g,* the plasmid pellet was washed with 70% ethanol and centrifuged for 5 min 12,000× *g*. The pellet was dried for 20 min at room temperature and the purified supercoiled pUC19 was eluted in TE buffer and transfer into a DNA LoBind tube. DNA concentration was calculated from the absorbance at 260 nm with a DNA calculator tool (from Molbiotools.com, © 2019 Vladimír Čermák, Prague, Czech Republic).

#### 2.2.2. Linear D-Loop

Synthetic oligonucleotides (sequences are listed in Appendix A, [25]) were purchased from Eurogentec (Liege, Belgium). To generate the linear D-loop substrate (S91/29/91), equal molar concentrations of oligonucleotides (Up1_91 nt, Up2_29 nt 5′Fam, Down_91 nt) were mixed in buffer containing 10 mM HEPES, pH 7.5, 100 mM NaCl and heated at 95 °C for 5 min, then were gradually cooled down to 20 °C. Then, the substrate was purified from a native acrylamide gel followed by DNA extraction and ethanol precipitation. As control, a primer/template substrate (S29/91, Appendix A) was prepared followed by the same annealing protocol without additional purification.

#### 2.2.3. Linear Primer/Template

Synthetic oligonucleotides (sequences are listed in Appendix A) were purchased from Eurogentec (Liege, Belgium). To generate substrates (S30/87/30), equal molar concentrations of oligonucleotides (Up1_30 nt 5′Cy5, Down_87 nt, Up2_30 nt) were mixed in buffer containing 10 mM HEPES, pH 7.5, 100 mM NaCl and heated at 95 °C for 5 min, then were gradually cooled down to 20 °C.

### 2.3. Enzymatic Assays

#### 2.3.1. D-Loop Formation Assays

An amount of 25 nM 5′HL 675 L93 was first incubated with increased quantity of RadA (0, 0.4, 0.8, 1.6, 3.2, 6.4 µM) for 10 min at 65 °C in a buffer containing 20 mM Tris-HCl, pH 8.0, 10 mM DTT, 50 µg/mL BSA, 10 mM MgCl_2_ and 2.5 mM ATP (when indicated). Then 25 nM of purified supercoiled pUC19 was added and incubated for another 10 min. The final reaction volume was 20 µL containing 125 mM NaCl. Reaction was stopped by addition of 5 µL of proteinase K (250 µg/mL), 2.5% SDS, 200 mM EDTA and incubated at 37 °C for 15 min. The samples were mixed with equal volume of 20% Ficoll and subjected to a 1.2% agarose gel electrophoresis in 1× TAE buffer. The products were visualized by Typhoon FLA 9500 (GE Healthcare, Chicago, MI, USA) and quantified with ImageQuant software. RadA-dependent D-loop (%) was calculated with the densitometry measurement of formed D-loop as a percentage of total lane densitometry after data normalization and subtraction of the D-loop background from the control without protein. Experiments were performed in triplicate.

#### 2.3.2. Circular D-Loop Extension Assays

5′Fam L93 (25 nM) was first incubated with 0.8 µM RadA for 10 min at 65 °C in a buffer containing 20 mM Tris-HCl, pH 8.0, 10 mM DTT, 50 µg/mL BSA, 10 mM MgCl_2_, 0.2 mM dNTPs and 2.5 mM ATP (when indicated). Then, 25 nM of purified pUC19 was added and incubated for another 10 min. Next, the reaction was mixed with DNA polymerases at indicated concentration for a further 60 min. PCNA was added to the reaction mixture when indicated at 675 nM. The final reaction volume was 20 µL, containing 130 mM NaCl. DNA products were separated either by native agarose gel (as described in D-loop formation assays) or by denaturing gels. For denaturing gel electrophoresis, the reactions were terminated by addition of 86% deionized formamid, 0.01 N NaOH and 10 mM EDTA with equal volume. DNA products were analyzed by 5% denaturing polyacrylamide gel (8 M urea) or by 1% alkaline agarose gel. The electrophoresis images were obtained and the products were quantified as described above. The alkaline agarose gel was stained with SYBR Gold, scanned first to visualize HL-647-labeled products and then scanned for SYBR Gold for detecting the ladder (2-Log DNA ladders, New England Biolabs, Ipswich, MA, USA) and the plasmid pUC19.

#### 2.3.3. Linear D-Loop Extension Assays

5′Fam linear D-loop (S_91/29/91_ or S_29/91_) DNA (25 nM) was incubated with 225 nM DNA PolB or PolD at 65 °C in a buffer containing 20 mM Tris-HCl, pH 8.0, 10 mM DTT, 50 µg/mL BSA, 10 mM MgCl_2_ and 0.2 mM dNTPs. PCNA was added to the reaction mixture to 225 nM, when indicated. The final reaction volume was 20 µL, containing 130 mM NaCl. Reactions were terminated by addition of 20 µL of 86% deionized formamid, 0.01 N NaOH, 10 mM EDTA and 1 µM reverse complement DNA (Trap_60 nt, Appendix A). DNA products were analyzed after separation into an 8 M urea and 15% denaturing polyacrylamide gel and analyzed as described above.

#### 2.3.4. Strand Displacement Assays

5′Cy5 linear substrate (S_30/87/30_ or S_30/87_) DNA (25 nM) was incubated with 225 nM PolB or PolD (exo−) at 65 °C in a buffer containing 20 mM Tris-HCl, pH 8.0, 10 mM DTT, 50 µg/mL BSA, 10 mM MgCl_2_ and 0.2 mM dNTPs. The final reaction volume was 20 µL, containing contained 130 mM NaCl. Reactions were terminated by addition of 20 µL of 86% deionized formamid, 0.01 N NaOH, 10 mM EDTA and 1 µM reverse complement DNA (Trap_87nt, Appendix A). DNA products were analyzed as described above.

## 3. Results and Discussion

During the HR process, the crucial step of DNA synthesis allows faithful recovery of the missing genetic information. Strand invasion by RadA nucleoprotein filament forms a D-loop recombination intermediate that will serve as a primer for a DNA polymerase to initiate DNA synthesis. However, no study about biochemical assays with archaeal DNA polymerases in a DNA recombination context has been reported to date. Here, we reconstituted an in vitro strand synthesis assay system using plasmid-based D-loop substrate. With the aim to study the biochemical properties of archaeal DNA polymerases with or without their accessory proteins on these specific substrates, we combined strand exchange and DNA extension reactions using purified recombinant proteins of *P. abyssi* (RadA, PolB, PolD, PCNA) (Appendix A).

### 3.1. RadA Catalyzes D-Loop Formation on a Circular DNA Substrate

The RadA recombinase from *P. abyssi,* a member of the RecA family proteins, is a 38.8 kDa protein (Uniprot id: Q9V233). For the archaeal members of this family, in the presence of ATP or ADP, RadA multimerizes and forms a nucleoprotein filament with 3-nucleotides binding for each monomer [10,19,57]. After that, the recombinase performs a homology search to find a dsDNA having homologous sequence. The global mechanism is reviewed in [12,56,58]. Briefly, the homology recognition and short strand exchange require binding of ATP, allowing filaments to function as rotary motors. If the bound dsDNA is homologous to the ssDNA, strand exchange is performed through Watson–Crick base pairing and structural transition in the nucleoprotein filament regulates its sensitivity to the homology length [59]. In Bacteria, the physical distance between the two DNA-binding sites of RecA dictates a minimal homologous region of 8 nucleotides (nt) to initiate strand exchange [60]. Then, the nucleoprotein filament is disassembled depending on ATP hydrolysis and may be supported by mediators.

Here we purified the recombinant RadA protein without affinity tag, following a protocol used for *P. furiosus* RadA as described earlier [10,11]. As a control for RadA ssDNA binding activity, we used a fluorescent-labeled ssDNA (93 mer) in electrophoretic mobility shift assays (Appendix A). As described for *Pf*RadA, here *Pa*RadA bound ssDNA without requiring ATP and within range of protein in agreement with the stoichiometry described above. This ssDNA contained a homologous sequence to a portion of pUC19 plasmid, and was used as the strand to be exchanged by the RadA recombinase. For the plasmid-based D-loop assay, RadA was first incubated with the ssDNA to form the nucleoprotein filament, and then, provided pUC19 (dsDNA with 2686 bp) to the reaction mixture to trigger the homology search (Figure 1A). As mentioned previously by the biochemical studies with eukaryotic proteins [26,61], the purification level of the plasmid used as donor dsDNA is crucial to implement a robust and reproducible result of the in vitro recombination assays. Consequently, we added a final purification step where only the supercoiled form was collected and extracted after separation on gel electrophoresis (see materials and methods section) (Appendix A). This supercoiled form of pUC19 was used as the DNA template for the recombination assay.

The RadA-mediated nucleofilament formation induced D-loop formation on the pUC19 plasmid as shown in Figure 1B. The upper band with a lower migration velocity, by gel electrophoresis, appeared in the lanes 4 to 8 and should correspond to the ssDNA that invaded the complementary region of pUC19. The staining of the same native gel with SYBR Gold (Appendix A) confirmed the superposition of the 93 nt ssDNA signal (green) with the plasmid signal (red). This implies that a part of the 93 nt ssDNA invaded into the pUC19 plasmid and hybridized to the homologous region.

These products increased with the concentration of RadA and the reaction is clearly in an ATP-dependent manner (no product in lane 9 without ATP). It is generally known that the binding and hydrolyzing of ATP are required to achieve the strand exchange reaction by the RecA family recombinases, although the DNA binding of the recombinases is not ATP-dependent [11,62]. We noticed a faint band at the same position of the D-loop from the reactions without RadA (lane 3) and without ATP (lane 9). These bands are probably derived from the partial annealing of the ssDNA on melted zone of the plasmid, favored by the temperature of the assay at 65 °C.

For the reaction condition of this strand exchange, 0.8 µM of RadA is theoretically sufficient to cover 25 nM of the ssDNA with 93 nt from the stoichiometry of 1 monomeric RadA for 3 nucleotides. However, higher yields of D-loop were obtained with higher concentrations of RadA and the D-loop formation reached 33% as a maximum by 3.2 µM of RadA (Figure 1C). The apparently low rate of the strand exchange reaction by the archaeal RadA in vitro was similar to those observed by Rad51 on the same substrate (~30%) [26]. Increase in the reaction temperature did not improve the final D-loop yield (Appendix A).

Some regulators probably exist to enhance the strand exchange activity of RadA. Several RadA paralogs are supposed to regulate RadA activity and may support filament formation on RPA/SSB-coated strand in Archaea [11,14,63,64,65,66]. In *Sulfolobus solfataricus*, a recombinase paralog Ral3 stabilizes the RadA presynaptic filament [64] while Ral1, in conjunction with Ral2, was recently shown to negatively regulate RadA exchange activity [67]. In *P. abyssi*, only one candidate to be considered from the sequence homology was found. However, any function of this protein, called RadB, has not been identified. In *Haloferax volcanii*, RadB is supposed to induce conformational change in RadA to promote its polymerization on DNA [68]. In *P. furiosus,* a direct interaction between RadA and RadB has been detected [11], but functionally the binding of RadB on DNA negatively interfered with the formation of the RadA nucleoprotein filament. Here, in the absence of a stable *P. abyssi* RadB purified protein, we were not able to test whether RadB would help RadA in the D-loop formation assay.

Biochemical studies with eukaryotic proteins demonstrated that the ssDNA binding protein RPA, if added to the partially formed nucleoprotein filament, promotes the fully assembled Rad51 filament (reviewed in [56,69]). When added before Rad51, RPA binding affinity to ssDNA usually inhibits filament formation of Rad51 and Rad52 is required as a mediator on this RPA-coated ssDNA. During our experiments we also added *P. abyssi* RPA at different steps of the nucleoprotein filament formation reaction, but RPA always reduced D-loop formation in all conditions (data not shown). Taken together, the archaeal RadA recombinase probably acts in coordination with modulators that remain to be identified and characterized to catalyze efficient nucleofilament formation and strand exchange in vivo.

For the reconstitution of recombination-associated DNA Synthesis with eukaryotic proteins, it was reported that 30% of D-loop formation was sufficient to characterize the following step of DNA synthesis [26]. As we produced a similar amount of D-loop substrate (33%) in our archaeal in vitro system, we next sought to examine the DNA synthesis activity of the archaeal DNA polymerases on this substrate.

### 3.2. P. abyssi DNA Polymerases Extend RadA-Dependent D-Loops

D-loop formation provides a primer with a 3′ end available for further 5′ to 3′ DNA synthesis by DNA polymerases. In the next experiments we sought to find out which of the two DNA polymerases from *P. abyssi*, PolB and PolD, could use this substrate for DNA synthesis. We added PolB or PolD to the solution after RadA-mediated D-loop formation and incubated for further 60 min at 65 °C as described in Figure 2A. We used DNA polymerases with mutations to disrupt 3′-5′ exonuclease activity for this study just to simplify data analysis. Preliminary experiments with various concentrations of DNA polymerases were performed and selected 675 nM for further analysis (Appendix A). As shown in Figure 2B, addition of DNA polymerases led to accumulation of new DNA products with slower migration rate than the initial D-loop (compare lane 4 with lanes 5–6 and 8–9). These bands appeared only in presence of RadA, ATP and DNA polymerases. It was clear that the D-loop signal decreased (from 14% to 3%) balanced with increasing of the lower migrated signals especially in presence of PolB (from 0% to 30%), suggesting that the reaction products correspond to the signals of extended D-loop (Appendix A).

To further analyze the reaction products by DNA polymerases, the same reaction mixtures were separated on denaturing acrylamide gel. The Figure 2C shows DNA strands extended from the ssDNA that invaded into pUC19 with various lengths. The extended products were detected as some discrete sizes in Figure 2B, but they were separated to various sizes by the denaturing condition in Figure 2C.

Next, we evaluated whether PCNA, which interacts and confers processivity to PolB and PolD [32], would improve the extension reaction in the D-loop. We reported the direct interaction between PCNA and Mre11/Rad50, a sensor of DSBs previously [70], suggesting that PCNA is present on DNA during the resection process, the early step of DSB repair in *P. abyssi*. Therefore, PCNA is suspected to be on DNA for DNA synthesis during HR. As shown in Figure 2C, addition of PCNA increased the length of the extended product by both PolB and PolD as compared with the reactions by each DNA polymerase by itself. The alkaline denaturing agarose gel electrophoresis showed the difference of the product length more clearly (Figure 2D).

To confirm that the wild type PolB and D, named exo+, was also capable of robust D-loop extension, the same reactions were performed with exo+ enzymes (Appendix A). The similar images with slightly shorter sizes were obtained from the extension reaction from the D-loop structure, although some degradation products by the exonuclease activities were observed on free labeled-ssDNA (Appendix A).

PolB synthesized a longer product from the D-loop structure than PolD, but the sizes still did not reach the full-length of the plasmid template (Figure 2D, compare lane 2 to lane 7). These results suggest that DNA synthesis initiated at the D-loop is limited either by topological constraint or/and by additional partner missing.

In our recombination-associated DNA synthesis assay, extension of the 3′end could induce torsional strain in the closed circular template DNA depending on how D-loop extension proceeds [24]. In the expanding D-loop model, the D-loop size would grow along with DNA synthesis leading to the accumulation of topological constraints beyond the DNA polymerase. While in the migrating D-loop model, the D-loop bubble would stay with a constant size thanks to the release of the invading strand from the 5′ end [71].

In our assays, we observed in the native gel (Figure 2B), evenly spaced migrating species as well as majority of products (pointed out with arrows) both in the presence of PolB and PolD. These profiles are similar to those observed in the recombination-associated DNA synthesis assay using eukaryotic proteins with the same primer/pUC19 plasmid as ours [26]. In the eukaryotic study, they combined native with denaturing two-dimensional gel electrophoresis and demonstrated first that these evenly spaced migrating species likely corresponded to distinct topoisomers of extended-relaxed D-loops. In addition, they showed that this pattern displayed stall regions relevant to a mechanism whereby the D-loop size is directly correlated to the length of DNA synthesis, resulting in an expanding D-loop until DNA synthesis stalling. In their assays, the initial stall region was relieved after addition of the Topoisomerase I which suppressed topological constraints. However, no significant difference was observed in distribution of the product length, suggesting that the DNA synthesis process would follow a migration D-loop model for longer DNA products (>200 nt). We obtained the similar migration profiles of extended D-loops by the *P. abyssi* DNA polymerases, and therefore, suggest that Archaea has a similar mechanism to extend DNA strand in the D-loop structures. Additional experiments with topoisomerases would be required to confirm the hypothesis and get a better understanding of the mechanism in the archaeal cells.

To summarize, on these assays we reconstituted an in vitro system showing that RadA provided a D-loop substrate usable for further DNA extension by both archaeal DNA polymerases, PolB and PolD. PCNA supported DNA polymerase activity as the processive factor to go further in extending DNA. However, PolD was less efficient to extend DNA at the D-loop as compared with PolB. PolB managed to extend 88% of total ssDNA primer engaged in D-loops compared to 46% for PolD (Figure 2B, compare lanes 5 and 8). This significant difference was also noticed with PolD exo+ (Appendix A). To try to better understand this functional difference, we next compared PolB and PolD activities on simpler D-loop substrates.

### 3.3. PolD is Less Efficient Compare to PolB to Extend D-loop Like Substrate

We made a D-loop substrate by in vitro recombinase activity of RadA, and therefore, we could not thus conclude if the presence of RadA onto DNA would not affect the DNA synthesis activities of PolB and PolD. One report showed that the RecA nucleoprotein filament was required for activation of the Pol V-catalyzed TLS in Bacteria [72]. No study has been reported showing physical or functional interaction of the RadA recombinase with either PolB or PolD to date. In our hands, any of the purified *P. abyssi* RadA, PolB, and PolD were not co-immunoprecipitated (data not shown).

To examine DNA polymerase activity on recombination intermediate in absence of RadA, we used a synthetic linear D-loop substrate as a model substrate for the following DNA synthesis. The synthetic D-loop contains a 5′ fluorescent 29-nucleotide primer that can be extended by strand displacement of the 30-base pair duplex to form a 60-nucleotide-long product. The polymerase can extend the primer by 1 nt before encountering the dsDNA region of the D-Loop substrate (Figure 3A).

We first tested different concentrations of PolB and PolD in a standard experiment, and selected the concentration of 225 nM for further kinetic analysis; both polymerases behaved similarly in synthesizing the full-length product (Appendix A). Then, we compared the DNA synthesis of PolB and PolD on two DNA substrates, the linear D-loop and a simple primed substrate (Figure 3A). We found a clear delay of extension by PolD as compared with PolB to fully extend from the D-loop (Figure 3A, left panel). Similar delay was observed when comparing PolD exo+ and PolB exo+ (data not shown). It was not clear if the difference of efficiency between PolB and PolD was related to primer recognition or strand displacement activity.

To examine the strand-displacement activity in our reaction conditions, we used another synthetic substrate, in which a 5′ labeled 30 nt is annealed to one extremity of the 87 nt template, whereas another 30 nt oligonucleotide is hybridized at the other end resulting in a 27 nt ssDNA gap, as illustrated in Figure 3B. As this substrate was not purified, an excess of the labeled primer may appear and remained after extension reaction. The reaction products using this substrate were analyzed by a denaturing gel, and found that PolB and PolD were able to fully extend the primer to the end of the template DNA (87 mer) with a stall at 57 nt corresponding to the ss/dsDNA junction, indicating that PolB and PolD have shome DNA strand displacement activity. However, it was clear that PolD extended less efficiently than PolB after reaching the ss/dsDNA junction.

For primer recognition, a gel retardation assay was performed to compare the binding affinities of PolB and PolD to the D-loop substrate. However, this experiment did not allow us to determine if PolD has a reduced affinity for D-loop substrate as compared to PolB (data not shown). Another possibility is that access of PolD to the primed substrate might be limited before displacing DNA strand, because the space around the 3′-terminus of the primer in this substrate is not wide enough for PolD, which is a complex of 215 kDa with a predicted Rg of 39.1 Å (calculated from the structure 6T8H, [73]) (Appendix A). In comparison, PolB, a relatively small molecule with a molecular weight of 89 kDa and a predicted Rg of 28.5 Å (calculated from the structure 4FLU, [33]), is easy to access to the primer region. On the D-loop substrate, the access to 3′end of the primer should be harder by DNA topology constraint especially in the context of a supercoiled DNA template. We can reasonably suspect that protein partners such as a topoisomerase or even a DNA helicase could be recruited during the process to facilitate DNA polymerase work. Nevertheless, our current in vitro study demonstrates a consistent result that PolB seems to be more suitable DNA polymerase to work on a D-loop substrate.

## 4. Conclusions

In conclusion, we reconstituted an in vitro system for recombination-associated DNA synthesis process with the recombinant proteins of the hyperthermophilic archaeon, *P. abyss*i: RadA, PCNA, PolB, PolD. Based on our results, we propose that both DNA polymerases can interact and extend the recombination intermediate provided by RadA. In addition, PCNA loading stimulated DNA polymerases to displace the DNA strand and thus further extend the primer. Nevertheless, we suspect that the difference in strand displacement ability accounts for the difference in efficiency for the extension of D-loop substrate and that structure of PolD could also be disadvantage to get access to D-loop primer. This last observation could be consistent with previous genetic studies. Gene deletion studies in *Thermococcus kodakarensis*, and *Methanococcus maripaludis* showed that only PolD was essential for cell survival and may be the only replicative DNA polymerase required to replicate both the leading and lagging strand [50,51]. Interestingly the recent study demonstrated that a strain deleted of PolB has higher sensitivity to DNA-damaging agents such as gamma-ray irradiation, main source of double-strand breaks [49]. Together with these data, we propose that PolB could be a good candidate for DNA synthesis during HR process in our archaeal model, and that PolD could take over if required.

These data are noteworthy in respect to the potential role of HR proteins in the replication initiation. In the process named Recombination-Dependent Replication (RDR), the replication machinery is assembled onto D-loop recombination intermediates and the invaded 3′DNA end is used as primer for leading strand synthesis (reviewed in [74]). As already mentioned in the introduction, genetic studies in Archaea showed that *oriCs* were not always essential for growth and were not always systematically activated [8,9]. Interestingly in these studies, the *oriC*-independent replications are related to the HR process.

In the RDR process, beyond the repair of a DNA strand, full genetic information has to be replicated. As shown in our present study, archaeal PolB and PolD, associated with PCNA, have the capability to extend the primer provided during the strand exchange by RadA. Biochemically, it would be interesting to further analyze the archaeal replisome machinery assembling onto a D-loop substrate. PolD, as an essential DNA polymerase in several Thermococcales species, would have a major role in such a context. However, we could imagine a potential cooperation with PolB which has been shown to be the more suitable for HR process. Currently, *T. barophilus* would be a model organism for genetic studies. The *polB* deletion mutant has been isolated and isolation of the *cdc6* deletion mutant is now underway. These materials would be useful for further understanding of interaction between recombination and replication processes in Archaea.

## Figures and Tables

**Figure 1 biomolecules-10-01045-f001:**
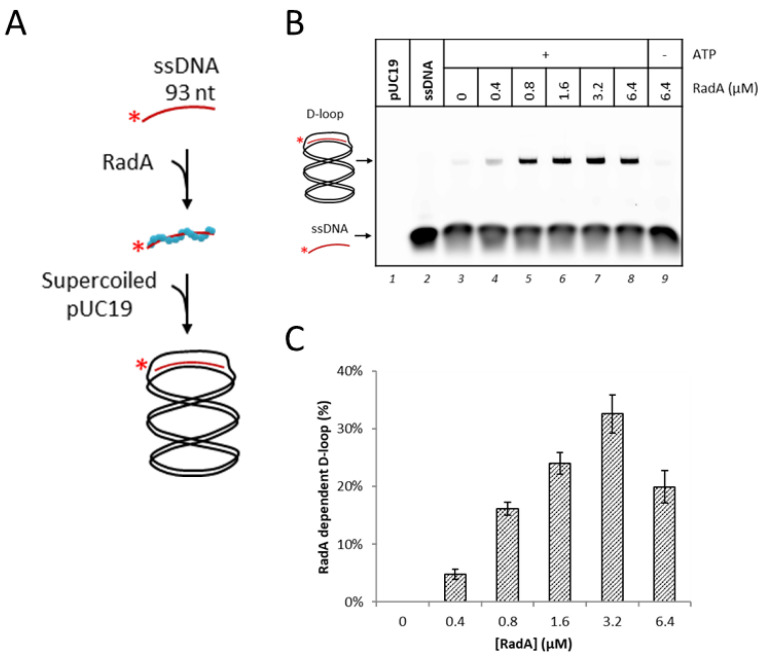
*P. abyssi* RadA recombinase activity catalyzed displacement-loop (D-loop) formation. (**A**) Schematic representation for the D-loop formation assay. Labeled linear ssDNA (93 nt) was incubated first with RadA to form nucleoprotein filaments before adding the purified supercoiled plasmid pUC19 for further homology search. (**B**) D-loop formation assay with increased quantity of RadA. An amount of 25 nM of labeled ssDNA (93 nt) was incubated with RadA for 10 min at 65 °C. Then, 25 nM of purified supercoiled pUC19 was added and incubated for another 10 min. DNA products were separated on a 1.2% native agarose gel and visualized by fluorescence. (**C**) Histogram representation of the D-loop formation assays for a range of RadA as observed in (**B**). RadA-dependent D-loop (%), densitometry measurement of formed D-loop as a percentage of total lane densitometry after data normalization and the D-loop background from lane 3 was subtracted. Experiments were performed in triplicate.

**Figure 2 biomolecules-10-01045-f002:**
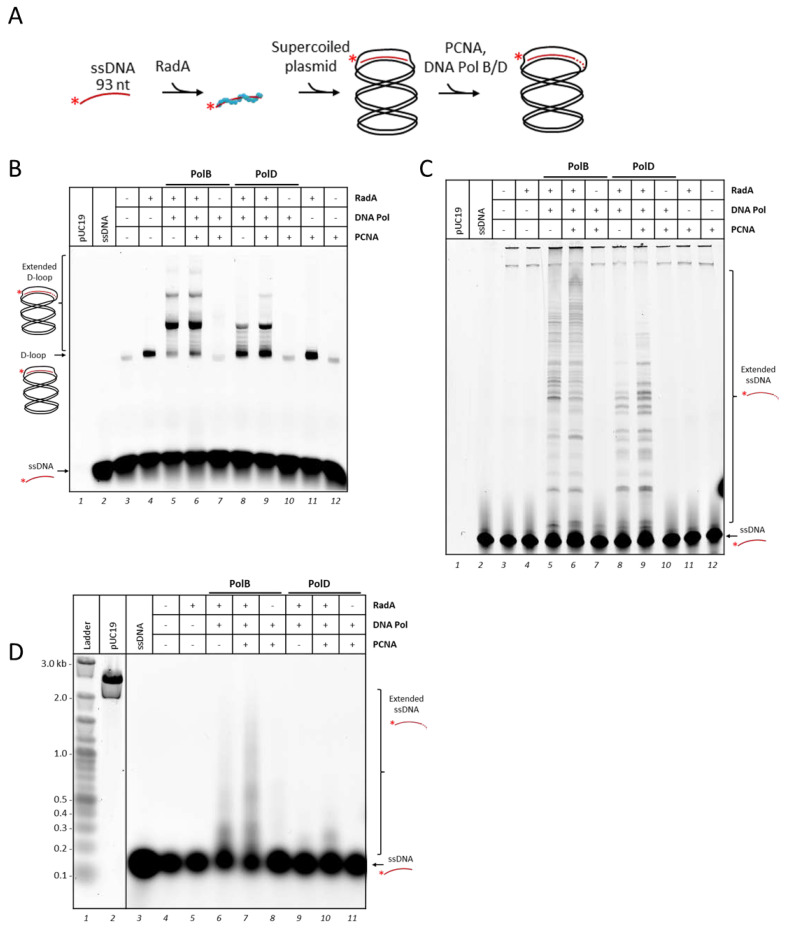
Addition of PCNA stimulates DNA extension by DNA polymerases on recombination intermediates. (**A**) Schematic representation for DNA extension by family-B polymerase (PolB) or family-D polymerase (PolD) following the D-loop formation by RadA described in Figure 1A. (**B**) Recombination-associated DNA synthesis assay. An amount of 25 nM of labeled ssDNA was first incubated with 1.6 µM RadA for 10 min at 65 °C. Then, 25 nM of purified supercoiled pUC19 was added and incubated for another 10 min. D-loop provided by RadA strand exchange activity was extended by 675 nM of PolB or PolD for 1 hr at 65 °C. DNA products were separated on a 1.2% native agarose gel. Same DNA products from (**B**) were separated as well in 5% denaturing acrylamide gel (**C**) or 1% denaturing alkaline agarose gel (**D**). When indicated, 675 nM of PCNA was added together with DNA polymerases. DNA products were revealed by fluorescence for HiLyte^TM^ 647 labeled DNA. The denaturing alkaline agarose gel (**D**) was also stained by SYBR Gold to detect the DNA ladder (lane 1) and pUC19 plasmid (lane 2). For all the experiments, controls were treated as the assays (volume and incubation time), when a protein was absent it was replaced by the corresponding buffer. The two bands at the top of the gel in lanes 3 to 12 are non-specific products corresponding to incomplete denaturation of pUC19 plasmid with residual labelled ssDNA fixed on melted regions.

**Figure 3 biomolecules-10-01045-f003:**
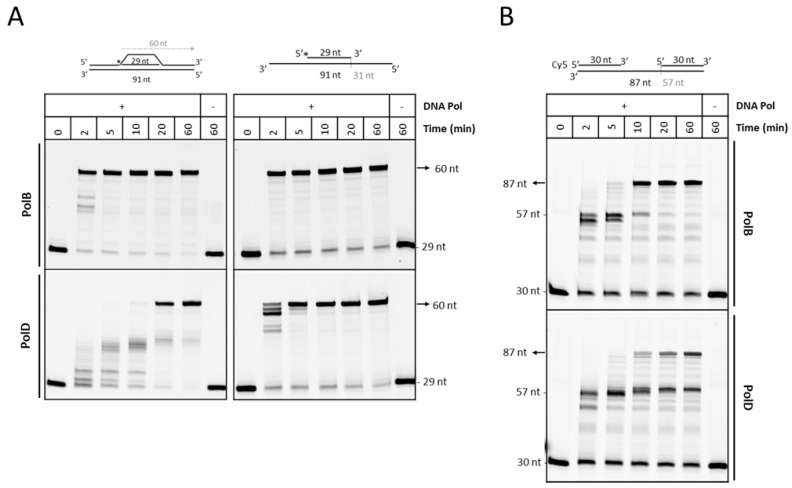
DNA synthesis activity of PolB and PolD on linear substrates. (**A**) Kinetics of reaction for extension of the 5′end labeled 29 nt for the linear D-loop S91/29/30 or primer/template S29/91. (**B**) Strand displacement activity of DNA polymerases. A quantity of 225 nM of PolB or PolD was incubated with 25 nM synthetic linear DNA substrates at 65 °C for a range of time indicated. DNA products were separated by gel electrophoresis onto a 15% denaturing acrylamide gel and revealed by fluorescence.

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
