# Peer review of "Role of RadA and DNA Polymerases in Recombination-Associated DNA Synthesis in Hyperthermophilic Archaea"

_biomolecules, 2020, doi:10.3390/biom10071045_

Round 1

Reviewer 1 Report

The faithful and accurate transmission of the genome through successive cell divisions requires a precise network of pathways coordinating, among others, DNA replication with DNA repair and recombination. Homologous recombination plays a central role in DNA repair and in the restart of stalled DNA replication forks. Interestingly, the key actors of homologous recombination in Archaea appear to be essential for survival in hyperthermophilic archaea. However, understanding how the advancing replisome deals with obstacles, such as DNA damage, and how coordination is achieved between the moving replisome and DNA repair components during replication-coupled DNA repair remains unclear. In the study by Hogrel et al, an in vitro assay for recombination-associated DNA synthesis has been reconstituted. In presence of the RadA recombinase, both PolB and PolD can take over RadA-mediated recombination intermediates. These results support an important role for PolB in homologous recombination but also identify PolD as a possible back-up polymerase. This observation is important as the division of labour between the two main DNA polymerases families in Archaea, PolB and PolD, is not established with the same degree of certitude as in the other domains of life. Yet, this study provides novel insights on the interplay between recombination and replication in hyperthermophilic archaea, and it should definitely be published in biomolecules, after minor revisions. Please find below a few points that the authors should consider when preparing the revised version of their manuscript.

Major point:

  • Space filling in the main figures (especially figure 2) is suboptimal. The schematic diagrams for the recombinase and polymerase extension assays in Fig1A and Fig2A are very useful but two small and difficult to read. The authors may consider rearrange the panels of their figures in order to increase the size of panels 1A and 2A.

Minor points:

  • The authors use either “PolD or PolB” or “the PolD and the PolB”. Please homogenize.
  • 2 – l. 53-54: … have revealed that the dynamic … is dependent on …
  • 2 – l. 75: In our … could be … Our ?
  • 2 – l. 79: … are processive enzymes
  • 5 – l. 196: Twenty-five nM … could be 25 nM
  • 6 – l. 278: … “increased with increasing” … please revise
  • 7 – l. 293-300: Quantitation of the percentage of D-loop formed as the ratio of D-loop signal on the total DNA signal in triplicate assays. The sentence is unclear please revise
  • 7 – l. 320: to be characterized … or … to be identified ?

Reviewer 2 Report

The manuscript entitled “Role of RadA and DNA polymerases in recombination-associated DNA synthesis in hyperthermophilic archaea” by Hogrel et al describes the nucleotide extension function of P. abyssi PolB and PolD polymerases in the context of strand invasion mediated by the RadA recombinase. Through a series of in vitro experiments, the authors determine that PolB is more efficient than PolD at extending the substrate within the invaded structure, further connecting PolB function with DNA repair activities.

Comments:

  • This appears to be the first report of recombinant P. abyssi RadA protein. While it is likely to be highly similar to the RadA of P. furiosus, it remains a formal possibility that it has different biochemical attributes. Information about basic activities should be either provided or a reference for a publication about this protein given. Missing information includes ssDNA-dependent ATPase activity, temperature optimum, and especially stoichiometry with ssDNA (nucleotides). This is critical information needed to understand the choices of protein concentration used in all the assays throughout this work and evaluate the results. If this is not previously published, the authors should show assays that demonstrate these basic biochemical activities.
  • Line 103: states the gene was optimized for expression. How? Please provide the genbank ID for the gene and note the changes.
  • Line 136: please provide the genbank ID number for PolB (and also for PolD, line 147)
  • Line 145: add the extinction coefficient and molecular weight for this (and all proteins throughout manuscript)
  • Line 154-160: Purification details (buffers, etc.) are missing. If this is previously published, please provide a reference. Otherwise, please add details sufficient for another lab to replicate the purification (more like the RadA purification section, which is very detailed and helpful).
  • Line 163: how homologous is the oligonucleotide to pUC19? Which section of pUC19 is it homologous to?
  • Line 169: It is known that use of alkalai lysis with a Quiagen kit to purify plasmids for use in D-loop assays results in denatured “bubble” regions. It is commendable that the authors worked to purify only supercoiled plasmid, but it is extremely likely that there are bubbles within their substrates. This is not a fatal problem, but the authors should be aware that the faint background bands in their results are probably due to these bubbles rather than exclusively temperature issues. In future experiments, they should consider using purification protocols that avoid alkalai lysis for their plasmids (sucrose gradients, etc.)
  • Line 196: the stoichiometry of RadA to nucleotides for this specific protein should be indicated.
  • Line 249: the references in this line include only one Pyrococcal species. The phrasing of this sentence and the one before it suggests that all these references are for Pyrococcus. This is confusing to the reader and should be rephrased for clarity.
  • Line 279: there is a faint product in lane 9. This is probably background produced with a bubbled plasmid. The intensity of this band should be subtracted from the other bands to account for background in the quantitation.
  • Line 289: D-loop assays measure strand invasion, not exchange. Low levels of strand invasion by the P. abyssi RadA could be the result of the low temperature, since 65C is significantly below the growth temperature of the organism. Here, knowing the optimal temperature for RadA activity would be helpful.
  • Line 303: in some archaea, the single-stranded DNA binding protein is called SSB instead of RPA.
  • Line 315: it is confusing when the narrative jumps from archaea to eukaryotes. This section should be clarified to make it obvious when the discussion is about eukaryotes instead of archaea.
  • Line 321: why is RPA required in conjunction with Rad51? Is the protein less efficient alone? Please expand and explain.
  • Line 335: active RadA is mentioned. Was inactive (or mutant) RadA also checked? Clarify.
  • Line 337: this comparison needs quantification.
  • Comment about the deproteinization step in the D-loop assays: it is possible that some of the observed bands on the gels are the result of gel shifts. Deproteinization with proteinase K at 37C is not always complete, especially with thermophilic proteins. A control experiment to verify that the reactions are completely deproteinized should be done. This could be a single reaction condition treated with proteinase K at 37C compared to 65C (proteinase K is active at this temperature) to verify that the bands are equivalent between the two conditions.
  • Figure 2: is the RadA only lane for each panel incubated at 65C for only 20 minutes (10 min with ssDNA then 10 min more after pUC19 addition) or is it incubated for the extended time used following PolB/D PCNA addition?
  • Figure 2C: What are the two bands at the top of the gel above the bracket on the right?
  • Figure 2 legend: please mention the purpose of the arrows in the legend.
  • Figure S1: why is the RadA protein running small on the gel? The text says it’s 38.8 kDa but it’s less than 37 kDa on the gel.

Reviewer 3 Report

Homologous recombination (HR) plays essential roles in both DNA repair and replication in archaea. DNA extension by polymerases after D-loop formation is one of the key steps in HR. In this manuscript, the authors investigated which DNA polymerases, PolB or PolD, was potentially the main player DNA extension in the hyperthermophilc eurychaeon Pyrococcus abyssi by in vitro reconstitution experiments. DNA extension by PolB and PolD on a circular substrate and several linear substrates catalyzed by the recombinase RadA was assayed together with associated factors PCNA and RPA. The results showed that both polymerases were able to take charge of processing the RadA-mediated recombination intermediates. But PolD was far less efficient, as compared with PolB, to extend the invaded DNA at D-loop substrate. PCNA stimulated the extension by both PolB and PolD, while RPA inhibited the D-loop formation. Because RadB, a potential accessory protein for RadA, was unavailable, its role in D-loop formation and DNA extension was not tested. The authors proposed that PolB could be a good candidate for DNA synthesis during HR process in their archaeal model, and that PolD could take over if required.

This is an interesting and well designed study, filling the gap in our understanding of HR in Euryarchaeota and will be helpful for further revealing of the mechanism of Recombination-Dependent Replication (RDR) in Archaea. The manuscript was elegantly written and I would suggest it to be published in the journal of Biomolecules without further revision.

Author Response

We are grateful to the referee for the reviewing of this manuscript.

The reviewer did not make any comments or mention any revision points.

Round 2

Reviewer 2 Report

.